# Relationship between Executive Functions and Creativity in Children and Adolescents: A Systematic Review

**DOI:** 10.3390/children10061002

**Published:** 2023-06-02

**Authors:** Tania Pasarín-Lavín, Amanda Abín, Trinidad García, Celestino Rodríguez

**Affiliations:** Department of Psychology, University of Oviedo, 33003 Oviedo, Spain; pasarintania@uniovi.es (T.P.-L.); abinamanda@uniovi.es (A.A.); garciatrinidad@uniovi.es (T.G.)

**Keywords:** creativity, executive functions, intelligence, working memory, inhibition, switching

## Abstract

(1) Background: Executive functions and creativity could play an important role in children’s education. To date, research on the relationship between these constructs has focused on adults. The objective of this systematic review was to analyze the relationship between executive functions (EFs) and creativity in children to provide teachers with tools to improve students’ abilities. (2) Methods: A total of 12 studies were identified using WOS, SCOPUS and PsycINFO, which matched the following criteria: (i) empirical studies with measures of executive functions and creativity; (ii) a sample of children or adolescents (3 to 18 years old); and (iii) in the previous decade (2012–2021). (3) Results: The results indicated a clear relationship between flexibility and creativity. Flexibility is positively correlated and inhibition is negatively correlated with creativity. There is no clear evidence that the remaining EFs, such as working memory, correlate with creativity. There was insufficient evidence on the relationship between intelligence, executive functions and creativity in a sample of children for the results to be generalized. (4) Conclusion: Future studies should consider the variability of standardized tests that measure these two constructs in order to be able to compare measurements and obtain generalizable results.

## 1. Introduction

The relationship between executive functions (EFs) and creativity has been thoroughly studied in samples of adults [1,2,3,4]. Many of our higher cognitive processes are put into action for us to be creative: working memory, flexibility, planning and inhibition, among others. This is why it is essential to understand the importance of EFs in creativity—taking into account the mediating value of intelligence, since it has been widely investigated in relation to creativity [5,6]. Over recent years, creativity and EFs have been highly valued and sought-after constructs in society [7], and scientific evidence shows that EFs play an important role in adult creativity [8,9]. The present study aims to analyze this relationship in a sample of children and adolescents.

### 1.1. Executive Functions (EFs)

EFs refer to higher mental processes allowing flexible and complex functions that direct behavior towards a goal [10]. In general, inhibition, working memory and shifting are considered the main executive processes [11] that other processes, such as planning, problem solving, etc., depend on [12].

It is well known that EFs present a common but independent variance factor, meaning that there is no perfect correlation between them [11]. This is why the three processes of working memory, inhibition and shifting should be included when evaluating EFs. Inhibition is the ability to inhibit or control automatic responses; working memory is the ability for temporary storage and processing of information; and shifting is the ability to unconsciously shift attention from one task to another [11,13].

Proper development of these EFs makes daily life easier. Additionally, a deficit in some of these functions is key to the diagnosis of some educational needs, such as ADHD [14], so much so that authors such as Filippetti and Richaud [15] claim that the development of EFs improves academic performance. This observation makes it important to know how they relate to other constructs, such as creativity and intelligence.

### 1.2. Creativity

Like with EFs, several authors [16,17,18] related creativity to an improvement in academic performance. In fact, creativity is one of the most widely-demanded skills in modern society in order to pursue a professional career [19].

The theoretical models that explain creativity begin with Guilford [17], who explained that creativity is made up of five components: (1) sensitivity, as the ability to quickly detect problems in order to solve them; (2) fluency, as the ability to produce a large number of ideas, words or associations; (3) flexibility, as the ability to switch from one idea to another, from one context to another, and to give varied responses; (4) elaboration, as the ability to perceive deficiencies, generate ideas and refine them to obtain new and improved versions; and (5) originality, as the ability to produce unusual ideas or responses. From Guilford, we move on to Csikszentmihalyi [20], who noted that creativity occurs in the interaction between a person’s thoughts and a sociocultural context considering three elements: (1) the domain, which consists of a set of rules in the culture of a society; (2) the field, which includes the individuals who give access to the field, for example, teachers; and (3) the person, who uses the rules of the field, interacts with the field and produces creativity.

Hence, creativity is a multifaceted construct which can be studied from various perspectives in the field of education. The two main forms of creativity are as follows: (1) divergent and convergent or (2) verbal and figural, although it can also be studied through its components, including fluency, flexibility, originality and elaboration [21]. However, we still need to know what makes certain people more creative than others. Alternatively, authors such as Rhodes [22] and Treffinger et al. [23] defined creativity with four Ps: person, product, process and pressure-context.

### 1.3. Executive Functions (EFs), Creativity and Intelligence

EFs are mainly important in generating new ideas for which flexibility is key [24]. This means that EFs are seen as fundamental elements for the creative process, and, hence, must be studied jointly [25].

This relationship has been carefully analyzed in samples of adults by many authors. Some found statistically significant results in this relationship, especially with regard to EFs, such as inhibition, working memory and flexibility [26,27,28].

Other authors found positive relationships in some of these EFs, for example, between working memory and creativity measured with various tasks [1,29]. It has been suggested that creativity needs information retrieved from memory to build new ideas [29]. The literature also indicates evidence of a relationship between inhibition and creativity, explaining that a lack of cognitive control benefits creativity, specifically with fluency and flexibility but not with originality [30,31]. Moreover, creativity seems to be related to shifting because it requires flexibility of thought to produce new and different ideas [3,26].

Many studies have demonstrated relationships between EFs and creativity, although the vast majority agree that the relationship depends on the measures used for each variable.

Finally, the intelligence–executive functions and intelligence–creativity binomial have been studied in recent years. The first seems to show more support, since authors such as Frith et al., Karwowski et al., and Silvia [32,33,34] found that executive tests correlate significantly with the results in intelligence tests, which leads them to affirm that the administered executive tests constitute an excellent measure of general intelligence. The intelligence–creativity binomial is more controversial, and no agreement has been reached on whether this relationship exists, but the latest studies in adults [35,36] agree that the relationship between these two constructs is significant.

Taking this into account, in order to analyze the relationship between creativity and executive functions, it is important not to leave aside the intelligence variable. Some of these studies have also assessed intelligence as a mediating variable in this relationship [30,37]. Authors such as Benedek et al. [1] showed that working memory was found to explain a notable part of the shared variance between intelligence and creativity. Benedek et al. [30] examined whether the relationship between EFs and creativity was mediated by intelligence. They found that inhibition primarily promoted the fluency of ideas, whereas intelligence specifically promoted the originality of ideas. Other authors have explained that working memory is a predictor of individual differences in intelligence [38].

### 1.4. The Current Review

In summary, it could be said that these factors are essential for school and life. Executive functions and creativity prepare children and adolescents to adapt to unforeseen changes and challenges that may occur in the future of our society [39,40]. However, these constructs have been extensively studied separately and the studies that relate one to the other have focused on adults. This may be because it is easier to evaluate these skills in adults. As indicated above, several studies have shown a relationship between creativity and EFs in adults, and it is important to confirm whether the same occurs in younger people. The relationship with intelligence is currently not clear, however, nor is it clear whether it is a mediator in the relationship between EFs and creativity.

Therefore, we set out to review the empirical work investigating these relationships in samples of children and adolescents (from 3 to 18 years old). The study was limited to research published in the last 10 years (2012–2021). Importantly, the current review is expected to offer contributions to education, literature and future research. Additionally, the results are expected to help teachers and schools to work on more EF-focused strategies to develop creativity and ability in students.

More specifically, the following research questions guided this systematic review:

RQ1. What is the relationship between creativity and EFs in children?

RQ2. Is intelligence a mediating variable in this relationship?

## 2. Materials and Methods

Systematic reviews are solid syntheses of evidence that aim to bring together a complete description of the knowledge in a particular field of research in a single document [39,40]. To ensure the quality and rigor of a systematic review, the documentation supporting the review process should be complete and include the following considerations: formulation of questions, the definition of inclusion and exclusion criteria, the definition of the search formulation, etc. [41]. The PRISMA guidelines (Preferred Reporting Items for Systematic Reviews and Meta-Analyses) provide a checklist for reviewers on how to report a systematic review and allows the rigorous analysis of all the studies found [42].

### 2.1. Search Strategy

The literature search used multidisciplinary databases (WOS and SCOPUS) and one specific database (PsycINFO) in March 2022. The main objective was to analyze the relationship between creativity and EFs, and the keywords used for the search were as follows: (“Creativity” OR “creative thinking” OR “creative ability” OR “divergent thinking” OR “originality”) AND (“executive function*” OR “cognitive control” OR “executive control”). Moreover, a multi-method approach was used to improve the review by including two points of view: a quantitative view using methodological search tools through search equations in databases and a qualitative filtering of the references of the articles read that were not included in the previous databases.

### 2.2. Selection Criteria

The inclusion and exclusion criteria were defined to extend our knowledge about the relationship between EFs and creativity. To be included in this systematic review, articles had to meet the following criteria: (i) written in English or Spanish; (ii) included a sample of children or adolescents attending school (age range from 3 to 18 years old); (iii) empirical studies, with at least one measure of EFs and one of creativity; (iv) full-text available; (v) sufficient descriptive statistics (age, origin, gender, N); (vi) published in the previous ten years (2012–2021); and (vii) provides statistical and correlational data between EFs (flexibility, working memory, inhibition) and creativity.

In contrast, articles were excluded when they met the following criteria: (i) studies focusing on fMRI or neuroimaging; (ii) intervention studies focusing on pretest; (iii) studies with insufficient information describing sample (N, gender, age, origin); (iv) clinical studies of individuals with pathologies; (v) studies not indexed in JCR or SJR journals; and (vi) theoretical research, books, handbooks and all types of gray literature.

### 2.3. Data Extraction

Endnote was used as a data manager in the article selection process. The first selection phase was based on ordering and identifying the articles by title. In instances where the title did not clearly indicate the object of the study, we continued to the next phase, which consisted of reading the abstract. If there were still doubts about compliance with the inclusion criteria, we moved to the full reading in the eligibility phase. In the first selection phase, the authors of the present review analyzed the inclusion and exclusion criteria, and in the eligibility phase, all full texts were independently analyzed by two reviewers to ensure rigor and to ensure that inclusion criteria were met.

The following information was collected from each article: type of study, year, journal, author, title, country of the sample, aims, variables, sample and age range, study design, evaluation instruments used and summary of the results.

### 2.4. Study Selection

A total of 1357 studies were found in the database searched (561 from WOS, 466 from Scopus, and 330 from PsycINFO). Duplicates (*n* = 579) were removed, and 680 papers were excluded by title (*n* = 604) or abstract (*n* = 76). In total, 98 articles were selected for eligibility assessment but only 80 had the full text available for analysis to produce the final sample. Ultimately, a total of 11 studies met the inclusion criteria, and the references in those papers were analyzed to find additional studies. This produced 52 potentially relevant papers. Additionally, 32 duplicates and 15 papers that were already in the previous review (3 that met the inclusion criteria and 12 that were excluded previously) were excluded. Subsequently, 5 studies were selected by abstract to analyze by full-text, only 1 of which met selection criteria. Finally, a total of 12 articles (11 from databases and 1 from reference selection) were selected, as shown in the following flowchart (Figure 1) that complies with the PRISMA Declaration [43].

## 3. Results

### 3.1. Main Methodological Characteristics

We begin by describing the bibliometric properties of the 12 studies included (Table 1), the general characteristics (Table 2) and then provide details on the relationship between the two variables (EFs and creativity).

These 12 articles were published in 10 different journals, which are all indexed in JCR and/or SJR. The following table shows their general characteristics (Table 2).

Over the previous decade, research in relation to EFs and creativity in children has been very limited, but interest in the subject has increased in the last two years. In addition, certain relevant, interesting methodological characteristics help us understand the content of the selected articles (Table 3).

As the table indicates, childhood (7–12 years) has been examined most *(n* = 7), followed by early childhood (*n* = 3) and adolescence (*n* = 2). The 12 studies reported results from 6 countries, mostly the Netherlands (*n* = 4), Argentina (*n* = 3) and the USA (*n* = 2) and the remainder from Canada, China and Spain. The sample sizes varied, ranging from 64 to 289 students.

More than half of the studies (58.3%; *n* = 7) evaluated the three EFs that are considered fundamental (inhibition, working memory and shifting): 16.7% only looked at inhibition (*n* = 2); 8.3% investigated general EFs (*n* = 1); 8.3% explored inhibition and working memory (*n* = 1); 8.3% looked at attention (*n* = 1); and only 33.3% took into account the intelligence variable (*n* =4).

The studies reported using a variety of measurement approaches for the analysis of creativity: 41.7% evaluated verbal creativity (*n* = 5); 8.3% used figural or non-verbal creativity (*n* = 1); and 50% took into account both types of creativity (*n* = 6). The measures also varied in terms of the focus of analysis: 75.0% of the studies analyzed general creativity (*n* = 9), while fewer studies focused on creativity specific to some area, such as mathematical creativity (16.7%; *n* = 2) or reading and writing (8.3%; *n* = 1).

One important point is that all of the studies used performance and more than three-quarters of them used standardized tests to evaluate EFs (75%; *n* = 9) and creativity (83.3%; *n* = 10).

In general, analyzing the data indicates the large number of characteristics that encompass creativity and EFs, which explains the high variability in the measures used. EF and creativity are multifaceted constructs, and as such, there is no agreement on the elements that make them up. This division into different threads can cause difficulties when analyzing and comparing them.

### 3.2. Relationship between EFs and Creativity

The first research question concerns the extent to which EFs and creativity are related. As previously mentioned, the literature presents different points of view about the relationship between EFs and creativity. In total, 33.3% of the 12 papers argued that inhibition correlated significantly with creativity; 25% stated that working memory and inhibition correlated with creativity; 16.6% related shifting and inhibition to creativity; 16.6% reported that attention correlated with creativity; and 8.3% related general EFs with creativity. Finally, two of the studies argued that working memory and inhibition did not correlate with creativity, claiming that only shifting did so.

On the one hand, in a sample of children ranging from 8 to 13 years old, Filippetti and Krumm [44] showed that working memory and inhibition together correlated significantly with creativity because they correlate with flexibility, and flexibility correlates with creativity when measured using TTCT [55] and CREA [56]. Similarly, in a sample of children with a mean age of 10.01 years, Krumm et al. [9] reported that working memory, cognitive flexibility and inhibition measures with Stroop were significantly related to creativity measured in the same way. Chevalier et al. [54] reported that these same EF components also demonstrated a correlation with creativity in a sample of children aged 3 to 5 years old.

On the other hand, Sánchez Macías et al. [48] found positive, significant correlations between creativity measured with TTCT [55], flexibility and verbal inhibition but not with working memory in children aged 14–17; Zhao et al. [53] reported that creative thinking measured with TTCT [55] and RAT [57] was associated with working memory and inhibition but not with flexibility in a sample of children aged 12 to 15; additionally, Krumm et al. [47] found strong, significant correlations between creativity, which was measured using TTCT [55] and CREA [56], flexibility and inhibition, but it depended on intelligence and its control.

Other authors found that creativity only correlated with inhibition or attention. For example, de Chantal and Markovits [46] reported that inhibition was positively related to creative thinking measured with an ad hoc test based on three different problems in a group of children aged 3 to 5 years old; Bai et al. [45] found that attention had a significant effect on originality in a sample of 5–7-year-old children; therefore, it has a significant effect on a component of creativity measured with AUT [17]; van Dijk et al. [52] also showed positive effects of selective attention abilities on creativity measured with AUT [17] and TTCT [55] in a sample with a mean age of 11.07 years.

Taking EFs as a general construct, Vaisarova and Carlson [51] reported that EF significantly predicted fluency and originality, showing a negative association between EF and AUT creativity scores for preschoolers (Adaptation of Wallach and Kogan’s [58]) as well as scores in the adapted Montweiller and Taylor’s test [59]. Specifically, in mathematical creativity, significant correlations were also found with inhibition [49,50], and correlations were even found with creativity in the general domain measured with TTCT.

In turn, some of these authors claimed that working memory and inhibition were not related to creativity [45] and some reported only relating inhibition and flexibility, not the other EFs [48].

### 3.3. Intelligence as a Mediating Variable

The second research question concerns the mediating effect of intelligence on this relationship. Four studies evaluated intelligence, but only one explained the importance of the construct in EFs and creativity. In fact, authors such as Filippetti and Krum [44] stated that people with high creativity showed greater flexibility without necessarily exhibiting higher intelligence. In another study, Krumm et al. [47] found that executive functions like inhibition and flexibility acted as mediators between creativity and fluid intelligence.

## 4. Discussion

The purpose of our study was to investigate how EFs relate to creativity and how these relationships are mediated by intelligence.

### 4.1. Relationship between EFs and Creativity

First of all, the review convincingly demonstrated that there is a relationship between flexibility and creativity. Many authors [3,9,44,48] positively correlated flexibility and negatively correlated inhibition with creativity. These correlations occur because a person with high flexibility and low inhibition shows a high creative capacity. These results are on similar lines as previous studies [26,30], which found positive relationships between flexibility, inhibition and creativity, proposing flexibility as the central factor.

On the other hand, there seems to be insufficient evidence for a relationship between creativity and EFs such as working memory [45]. These results are consistent with Sharma and Babu [28], who noted that previous research, such as Roskos-Ewoldsen et al. [60], showed that TTCT [55] is not demanding on working memory, leading to these results.

On similar lines, authors such as Bai et al. [45] and van Dijk et al. [52] noted that attention plays an important role in creative capacity measured with AUT [17]. This effect of attention was negative and indicates that children with a lower level of attention produce more original responses. This is in line with the findings of a series of recent studies on the role of inhibition and attention, indicating that low inhibition and attention lead to higher creativity [27,37].

It is important to consider the variability in the tests used. Measures for creativity were relatively stable: TTCT [55], AUT [17] and CREA [56] were the most widely used. TTCT was used to assess figural creativity, AUT was used to assess verbal creativity and CREA was used to assess general creativity.

On the other hand, in EFs, there was a lot of variability in the tests, depending on the EF being evaluated (inhibition, flexibility or working memory). It is observed that only one study evaluates EFs in a general way with the Minnesota Executive Function Scale [61]. To measure the different EFs, we also find variability, but the most used tests are the Wisconsin Card Sorting Test [62] and Five-Point Test [63] to measure flexibility, the WISC-IV digits subtest [64] to measure working memory and the STROOP test [65] for response inhibition. This indicates that there may be room for standardization of tests to measure children’s EFs and creativity since the standardization of the measurement of these constructs would make it possible to generalize the results with greater rigor and reliability.

### 4.2. Intelligence as a Mediating Variable

All the studies analyzed in the present review used a measure of intelligence but only one examined what happens between intelligence, creativity and EFs. Intelligence seemed to play an important role in the samples of adults [1,26,37] but, in samples of children, there is not enough evidence to allow the results to be generalized. This means it is important to dedicate future efforts to studying this relationship in children and adolescents since most intelligence tests have a clear relationship with executive functions [66].

## 5. Conclusions

This study has important implications in the field of research on creativity and EFs, noting that creativity is not so much what children know (intelligence) but how they use that information, how they inhibit it and how flexible they are with it.

Society in the 21st century demands creative professionals who demonstrate leadership, planning and problem-solving skills. This is why, at an educational level, more and more active methodologies are used where creativity is worked on transversally [67]. This research supports the relationship between creativity and some EFs, and intends to promote future research to discover whether these constructs have a more specific relationship in children and adolescents, as is observed with a sample of adults.

The findings of this study indicate that educational interventions focused on cognitive training are needed to develop creative skills. This would also result in improvements in the students’ academic performance and in the development of skills they will need as future professionals. In practical terms, our results emphasize the importance of focusing on individual cognitive control, such as inhibition, flexibility and attention.

The present study has a number of limitations. The sample of articles was limited, although interest in this topic has increased in recent years. This indicates the importance of continuing to investigate the relationship between these two constructs. The measures of creativity tended to focus on divergent thinking and do not represent broader conceptions of creativity. This means that these measures may not be accurate and objective. Both creativity and executive functions are specific skills, meaning that they must be specifically measured. In addition, the variability of measures for executive functions does not allow a rigorous comparison between the studies since there were more than 30 different types of measures. Furthermore, the vast majority of the articles included in the study showed a trend towards positive results and there may be publication bias in this field. Lastly, in future studies, it could be important to analyze what happens with intelligence in relation to these two variables.

## Figures and Tables

**Figure 1 children-10-01002-f001:**
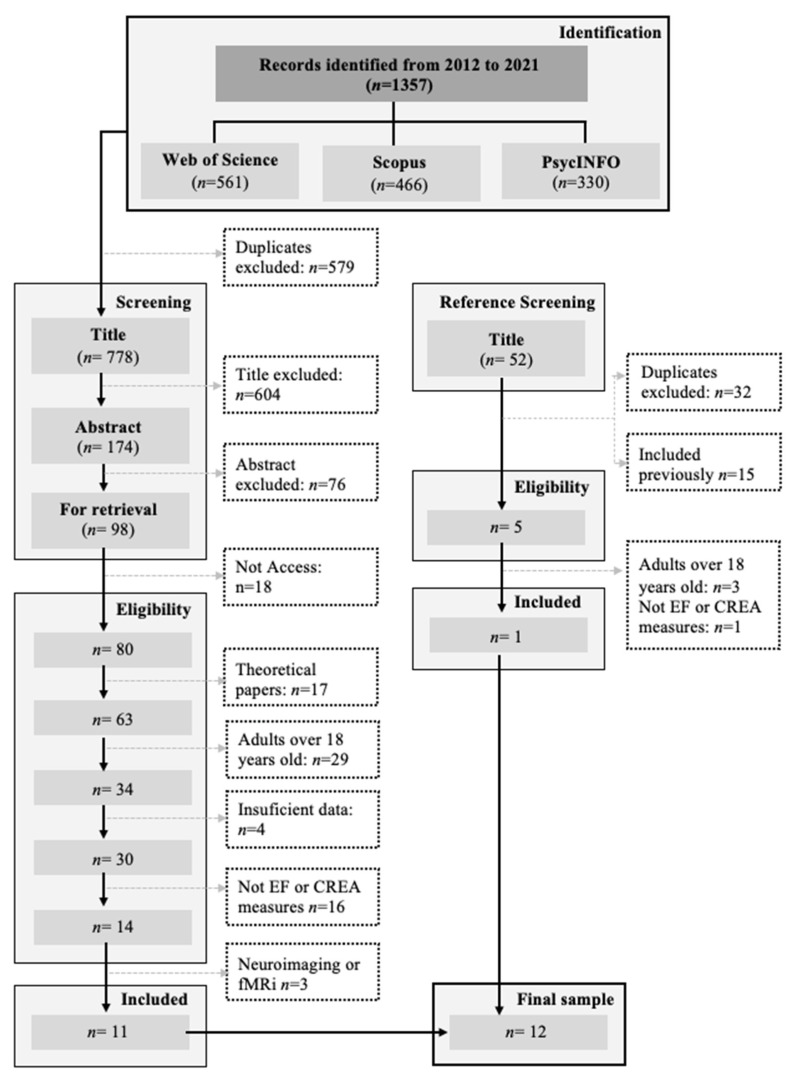
Flowchart based on PRISMA.

**Table 1 children-10-01002-t001:** Overview of the research on executive functions and creativity.

First Author (Year)	Sample			Measures	Outcomes
	Location	Size/Design	Age Range		
Filippetti et al. (2020) [44]	Argentina	**Study 1:** 112 children (M = 91.75; SD = 10.56) **Study 2:** 177 children (M = 9.94; SD = 1.24) **Descriptive design and Structural equation model (SEM)**	**Study 1:** 8–12 years **Study 2**: 8–13 years	**Study 1:** **Intelligence:** K-BIT; **Flexibility:** (1) Wisconsin Card Sorting Test; (2) Trail Making Test; (3) Five Point Test; (4) Semantic Verbal Fluency; (5) Phonological Verbal Fluency; **Working Memory:** Digits, Letters and numbers (WISC-IV). **Inhibition:** (1) Stroop color and Word Test; (2) NEPSY, Knock and Tap; (3) D2, Attention test (Spanish adaptation) **Reading and writing:** (1) Reading comprehension test (ENI); (2) PROESC writing test. **Study 2:** **Creativity:** (1) Figural Torrance Test (TTCT); (2) CREA; **Flexibility:** Same of numbers (1) and (3) in study 1; **Working memory:** Same of study 1; **Inhibition**: Same of number (1) in study 1; **Intelligence**: Same of study 1.	**Study 1:** Working Memory and Inhibition together contribute and support flexibility. This contribution depends on the tasks used and Inhibition contribution may depend on age. **Study 2:** There are consistent relationship between flexibility and creativity. Being creative requires a flexible thinking but depends on the tasks used and the type of flexibility measured (spontaneous or reactive).
Bai et al. (2021) [45]	Netherlands	102 children (M = 5.93; SD = 0.27) **Descriptive design and Multilevel regressions (longitudinal)**	5.45–6.53 years	**Divergent Thinking:** (1) Alternative Uses Task; **Inhibition**: (1) Programmed in E-Prime Go/NoGO task; (2) Animal Stroop task; **Shifting**: (1) Dimensional Change Car Sort (DCCS); (2) Animal shifting task; **Memory:** (1) Word Recall Backwards; **Selective Attention:** (1) Visual search task programmed in E-Prime (ad-hoc).	Executive functions such as attention and specific processes influence and are related to the creation of original ideas. The influence of inhibition and working memory cannot be confirmed, not predict/moderate originality.
De Chantal et al. (2017) [46]	Canada	**Study 1:** 32 children (M = 53.3 months) **Study 2:** 32 children (M = 47.06 months) **Correlational design and hierarchical regression**	**Study 1:** 41–64 months **Study 2:** 32–61 months	**Creativity:** (1) Generation task. Three different problems (ad-hoc); **Inhibition:** (1) DCCS task; **Reasoning**: (1) Logical reasoning task with a sets of problems (ad-hoc).	Inhibition plays an important role in children’s reasoning. The idea generation task is very similar to the creativity tests, so it can be said that creativity and some executive functions share mental processes.
Krumm et al. (2020) [9]	Argentina	200 children (M = 10.01; SD = 1.24) **Correlational design**	8–13 years	**Creativity:** (1) Figural Torrance Test (TTCT) (2) CREA; **Intelligence:** (1) K-BIT; **Working Memory:** (1) Digits, letters and numbers of WISC-IV; **Inhibiton:** (1) Stroop color-word test; **Shifting:** (1) Wisconsin Card Sorting Test Computer version; **Verbal Fluency:** (1) Semantic and phonological verbal test (fruits and animals); **Nonverbal Fluency (shifting):** (1) Five-Point Test; **Planning:** (1) Porteus Maze Test.	There are significant differences in the EF: (1) Working Memory would be involved in the search for creative ideas. (2) Inhibition (measured with Stroop) is positively related to creativity because it eliminates the interference of dominant responses. (3) Cognitive Flexibility is associated with fluency and creative flexibility and not with originality.
Krumm et al. (2018) [47]	Argentina	209 children aged (M = 9.96; SD = 1.23) **Correlational and SEM**	8–13 years	**Creativity:** (1) Figural Torrance Test (TTCT) (2) CREA; **Intelligence:** (1) K-BIT; **Working Memory:** (1) Digits, letters and numbers of WISC IV; **Inhibiton:** (1) Stroop color-word test; **Shifting:** (1) Wisconsin Card Sorting Test Computer version; **Verbal Fluency:** (1) Semantic and phonological verbal test (fruits and animals); **Nonverbal Fluency (shifting):** (1) Five-Point Test.	Positive correlation between creativity, flexibility, inhibition and intelligence. All executive functions correlate with creativity, but only inhibition and flexibility predicted creativity with intelligence as the mediating variable.
Sánchez-Macías et al. (2021) [48]	Spain	96 students (M = 14,5; SD = 0.85) **Correlational design**	14–17 years	**Creativity:** (1) Torrance Test of Creative Thinking (TTCT); **Working Memory:** (1) WISC-IV (Spanish adaptation; **Planning:** Hanoi Tower; **Inhibition:** (1) Stroop Test; (2) Go/noGO; **Shifting:** (1) Wisconsin Card Sorting Test; **Decision making:** Iowa Gambling task.	Positive correlation between creativity and flexibility and negative correlation between creativity and verbal inhibition. The correlations between creativity and the rest of the executive functions are not significant.
Stolte et al. (2020) [49]	Netherlands	278 children (M = 9.71; SD = 0.93) **Correlational design and SEM**	8–13 years	**Inhibition:** (1) Fish Game; **Shifting:** (1) Second block of the Fish Game; **Working Memory:** (1) Monkey Game to verbal updating; (2) Lion Game to visuo-spatial updating; **Mathematical Creativity (MC):** (1) MC Test Dutch translation; (2) Cito test; **General creativity:** (1) Test for Creativity Thinking Drawing production.	Positive correlation between creativity (general and specific) and working memory, but the role of inhibition and flexibility in creativity and mathematics cannot be generalized.
Stolte et al. (2019) [50]	Netherlands	**Fluency:** 80 participants (M = 9.95; SD = 0.84) **Flexibility:** 82 participants (M = 9.93; SD = 0.82) **Originality:** 81 participants (M = 9.96; SD = 0.82) **Correlational design and hierarchical multiple regression**	8–12 years	**Inhibition:** (1) Fish game; **Mathematical ability:** (1) Cito test; **Mathematical Creativity (MC):** (1) MC Test Dutch translation.	Results showed that mathematical creativity significantly correlated with components such as flexibility, fluency and originality. In the same way, that flexibility and originality significantly correlates with inhibition and this in turn produces a stronger relationship between mathematical ability and mathematical creativity of students.
Vaisarova et al. (2021) [51]	USA	**Experiment 1.** 103 children 53 4-year-old (M = 48.5 months)/50 6-year-old (M = 72.4 months) **Experiment 2.** To avoid age bias in Experiment 1, 78 5-year-old children with typical development from a university database were recruited (*M* = 66.7 months) **Comparative and Experimental design**	4–6 years	**Experiment 1.** **Executive Functions:** (1) Minnesota EF scale; **Effortful Control:** (1) Children’s Behavior Questionnaire (parents completed); **Creativity:** (1) Adaptation of Montweiler and Taylor’s; **Divergent Thinking:** (1) AUT for preschoolers (Adaptation of Wallach and Kogan’s); **Intelligence:** (1) Stanford-Binet Nonverbal Fluid Reasoning and Verbal Knowledge routing subtest. **Experiment 2.** **Excutive Functions:** (1) Same of experiment 1; (2) Flexible Item Selection Task (FIST) (3) Backward Digit Span; **Effortful Control:** (1) Same of experiment 1; **Intelligence:** Same of experiment 1; **Conformity:** (1) Adaptation of the Asch Social conformity paradigm; **Divergent Thinking:** Same of experiment 1.	Both studies showed negative relationships between EF and creativity. This suggests that cognitive skills do not enhance divergent and creative thinking and may take a backseat.
Van Dijk et al. (2020) [52]	Netherlands	70 (M = 11.07; SD = 0.69; 36 low stimulus; 34 high stimulus) **Correlational design**	9–12 years	**Creativity:** (1) Alternative Uses Task (visual) Computer task; (2) Semantic categories of the Torrance Test (TTCT); **Selective Attention:** (1) Subtest Sky Search of the Test of Everyday Attention For Children (TEA-Ch).	Overall positive effect of selective attention on creativity measures, especially originality. Attention causes distractors to be ignored, ignoring new ideas, but it helps to focus on details, which is why it scores more in originality.
Zhao et al. (2021) [53]	China	**Study 1.** 73 students (M = 13.8; SD = 0.6) **Study 2.** 68 students (M = 13.2; SD = 0.4) **Comparative and Experimental design**	**Study 1.** 13–15 years **Study 2.** 12–14 years	**Study 1.** **Working memory:** 4 tasks: 3 tasks to WM updating and 1 task WM span; **Inhibition:** (1) stroop task with Chinese character; (2) Flanker task; (3) GO/noGO; **Switching:** A digital test with a series of digits; **Creativity:** (1) Torrance Tests of Creative Thinking; (2) RAT (Chinese version. **Study 2.** **Pre- and Post-training Tests.** Working memory task, as described for Study 1, was used to assess near-transfer effects of the WM updating training; **Training Tasks.** Three adaptive running WM updating (a letter, animal, and visuospatial).	Creativity is related to working memory updating and not to inhibition, processing speed, working memory maintenance, or flexibility. This may be due to taking into account convergent thinking that other research does not value.
Chevalier et al. (2012) [54]	USA	250 preschool children (1) M = 3.71: age range = 3.67–3.83; (2) M = 4.45; age range = 4.42–4.5; (3) mean age 5.19; age range = 5.08–5.25 **Correlational design**	3 years 9 months, 4 years 6 months, and 5 years 3 months	**Flexibility:** (1) Shape School; **Inhibition:** (1) Go/No-Go task; **Working memory.** (1) The Nebraska Barnyard (adapted from the Noisy Book task).	Preschool-age children use working memory and inhibition to be flexible and creative. Students with greater inhibition are less flexible and concise, but this may be because at an early age it is more difficult to manage their cognitive abilities well.

**Table 2 children-10-01002-t002:** Systematic review sources: year, journal and language.

Variables	Frequencies	Percentage
**Year**		
2012	1	8.3%
2017	1	8.3%
2018	1	8.3%
2019	1	8.3%
2020	4	33.3%
2021	4	33.3%
**Journal**		
Child Neuropsychology	1	8.3%
Developmental psychology	1	8.3%
Journal of intelligence	3	25.0%
Memory and Cognition	1	8.3%
Personality and Individual Differences	1	8.3%
Psicogente	1	8.3%
Psychology of Aesthetics, Creativity and Arts	1	8.3%
Revista de Formación del Profesorado	1	8.3%
Thinking Skills and Creativity	1	8.3%
Trends in neuroscience and education	1	8.3%
**Language**		
English	10	83.3%
Spanish	2	16.7%

**Table 3 children-10-01002-t003:** Systematic review sources: sample age and country.

Variables	Frequencies	Percentage
Sample age		
Early childhood (0–6 years)	3	25.0%
Childhood (7–12 years)	7	53.8%
Adolescence (13–18 years)	2	23.1%
Country		
Argentina	3	25.0%
Canada	1	8.3%
China	1	8.3%
Netherlands	4	33.3%
Spain	1	8.3%
USA	2	16.7%

## Data Availability

Not appliable.

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
