# Peer review of "Relationship between Executive Functions and Creativity in Children and Adolescents: A Systematic Review"

_children, 2023, doi:10.3390/children10061002_

Round 1
Reviewer 1 Report
The paper focuses on a topic related to the relationships between executive functions and creativity in children. The study is based on a systematic review of studies and aims to systematize their findings in two directions. The first one focuses on the relationships between creativity and executive functions in children and the second explores the mediating role of creativity in this relationship. The selection criteria of the studies are clearly described and correspond to the research questions. The review demonstrates the relationship between creativity and flexibility and the insufficient evidence for a relationship between creativity and EFs such as working memory. The review reveals also that intelligence plays an important role in adults but there is not enough evidence to allow the results to be generalized for children.
The paper is very well written and I have a few suggestions to the authors. 1. The first suggestion is to describe in brief the main instruments measuring the studied constructs as it is mentioned in the paper they may limit the comparability of the selected studies. 2. The authors need to explain also the multi-method approach that “…was used to improve the review by including some additional relevant studies, taken from the references of the articles read, that were not included in the previous databases”. 3. The selected papers include studies with different design (experiments, tests, etc.). The authors should comment also how the design of the study and the research methods influence comparability of the results.
Author Response
Firstly, authors want to convey their sincere thanks to the editor and reviewers for their dedication and time spent on the review.
Below, the authors responded in detail to the proposed revisions. The changes are highlighted in the manuscript reviewed:
Reviewer 1:
The paper is very well written and I have a few suggestions to the authors.
- The first suggestion is to describe in brief the main instruments measuring the studied constructs as it is mentioned in the paper they may limit the comparability of the selected studies.
- Description was included in two final paragraphs of the second section of the discussion (page 13)
- The authors need to explain also the multi-method approach that “…was used to improve the review by including some additional relevant studies, taken from the references of the articles read, that were not included in the previous databases”.
- Method was clarified in page 3.
- The selected papers include studies with different design (experiments, tests, etc.). The authors should comment also how the design of the study and the research methods influence comparability of the results.
- Differences in research design are completed in Table 1 to clarify it, but we believe that the variability has more relation with the types of tests than with the type of design or method.

Reviewer 2 Report
The manuscript entitled "Relationship between executive functions and creativity in children. A systematic review" is about the relationships between executive function and creativity among children and adolescents. The introduction is well structured, but I recommend to the authors highlight the way EF and creativity are important in adolescents and children since the Authors only present the two constructs. On the methods, the Authors write their search strategies, data extraction, and selection criteria which help the reader to understand the data collection methods. The results are well presented. It is clear what the research is about. However, the discussion is short, and I feel that is missing the "big" conclusion of this study. What do you suggest after your finding? Another small issue: The title suggests that the study is about only children, but the research goal is aiming for adolescent well.
The english language is fine.
Author Response
Firstly, authors want to convey their sincere thanks to the editor and reviewers for their dedication and time spent on the review.
Below, the authors responded in detail to the proposed revisions. The changes are highlighted:
Reviewer 2:
The introduction is well structured, but I recommend to the authors highlight the way EF and creativity are important in adolescents and children since the Authors only present the two constructs.
- Information about the importance of EF and creativity in adolescents and children is clarified with evidence in page 3.
On the methods, the Authors write their search strategies, data extraction, and selection criteria which help the reader to understand the data collection methods. The results are well presented. It is clear what the research is about. However, the discussion is short, and I feel that is missing the "big" conclusion of this study. What do you suggest after your finding?
- New conclusions of this study was included in conclusions (page 14).
Another small issue: The title suggests that the study is about only children, but the research goal is aiming for adolescent well.
- The title was modified including adolescents.
